# Climate Impact of Idealized Winter Polar Mesospheric and Stratospheric Ozone Losses as caused by Energetic Particle Precipitation

Katharina Meraner[1] and Hauke Schmidt[1]

[1]Max Planck Institute for Meteorology, Bundesstraße 53, 20146 Hamburg, Germany

*Correspondence to:* Katharina Meraner (katharina.meraner@mpimet.mpg.de)

**Abstract.** Energetic particles enter the polar atmosphere and enhance the production of nitrogen oxides and hydrogen oxides in the winter stratosphere and mesosphere. Both components are powerful ozone destroyers. Recently, it has been inferred from observations that the direct effect of energetic particle precipitation (EPP) causes significant long-term mesospheric ozone variability. Satellites observe a decrease in mesospheric ozone by up to 34 % between EPP maximum and EPP minimum.
Stratospheric ozone decreases due to the indirect effect of EPP by about $10 - 15\%$ observed by satellite instruments. Here, we analyze the climate impact of winter boreal idealized polar mesospheric and polar stratospheric ozone losses as caused by EPP in the coupled climate model MPI-ESM. Using radiative transfer modeling, we find that the radiative forcing of a mesospheric ozone loss during polar night is small. Hence, climate effects of a mesospheric ozone loss due to energetic particles seem unlikely. A stratospheric ozone loss due to energetic particles warms the winter polar stratosphere and subsequently weakens the polar vortex. However, those changes are small, and few statistically significant changes in surface climate are found.

## 1 Introduction

Energetic particles enter the Earth's atmosphere near the magnetic poles altering the chemistry of the middle and upper atmosphere. Energetic particle precipitation (EPP) is the major source of nitrogen oxides ($NO_x$) and hydrogen oxides ($HO_x$) in the polar middle and upper atmosphere (Crutzen et al., 1975; Solomon et al., 1981). Both chemical components catalytically deplete ozone; $NO_x$ mainly below and $HO_x$ mainly above 45 km.

$HO_x$ is short-lived in the middle atmosphere and depletes ozone mainly in the mesosphere. In contrast, $NO_x$ persists up to several months in the polar winter middle atmosphere. Inside the polar vortex, $NO_x$ can be transported downward from the lower thermosphere to the stratosphere, where it depletes ozone (e.g., Funke et al., 2017; Sinnhuber et al., 2014; Hendrickx et al., 2015). Observational evidence of polar winter stratospheric ozone loss due to EPP is still limited. Only recently, long-term satellite observations with good temporal and spatial coverage became available. In austral polar winter EPP causes an ozone loss of about $10 - 15\%$ descending from 1 hPa in early winter to 10 hPa in late winter (Fytterer et al., 2015; Damiani et al., 2016). Extensive information on the current knowledge of energetic particle precipitation can be found in Sinnhuber et al. (2012) and Mironova et al. (2015).

Ozone loss influences stratospheric temperature and the polar vortex. The Northern Annual Mode (NAM) index is often used to describe the strength of the polar vortex, with positive NAM values indicating a strong polar vortex and negative NAM values indicating a weak polar vortex. Observations indicate that anomalous weather regimes associated with the NAM index can propagate from the stratosphere down to the surface (Baldwin and Dunkerton, 2001). Hence, energetic particle precipitation may provide a link from space weather to surface climate. Here, we study the impact of an ozone loss due to EPP on the circulation and subsequently on climate. Discussed are both a polar mesospheric and a polar stratospheric ozone loss.

Since the discovery of the ozone hole in the mid-1980s, the climate impact of a stratospheric ozone loss has been intensively studied (e.g., Shine, 1986; Randel and Wu, 1999; Lubis et al., 2016). Most studies concentrated on the climate impact of the ozone hole during austral spring and reported a cooling in the spring Southern Hemispheric stratosphere due to reduced absorption of solar radiation and a strengthening of the polar vortex. In contrast, our study concentrates on an ozone loss during the boreal polar night. During polar night reduced ozone slightly decreases the infrared cooling of the polar stratosphere resulting in a net (small) stratospheric warming (Graf et al., 1998; Langematz et al., 2003). However, both studies prescribed an ozone loss in the lower stratosphere.

Several studies suggested a significant influence of EPP on climate. Seppälä et al. (2013) and Lu et al. (2008) used reanalysis data to investigate the dependence of stratospheric temperature and zonal wind to the Ap-Index. They found a stratospheric warming up to $5 - 10\,\mathrm{K}$ for strong energetic particle precipitation descending from the stratopause to the mid-stratosphere. However, for the zonal wind response the two studies differ from each other. Seppälä et al. (2013) found a strengthening of the polar vortex, whereas Lu et al. (2008) showed a weakening of the polar vortex. Moreover, Seppälä et al. (2009) analyzed surface air temperature changes in reanalysis data for years with various strengths of EPP. They found a warming over Eurasia and a cooling over Greenland for winters with enhanced EPP, but could not rule out that the estimated changes are induced by NAM variability independent of EPP.

Other studies relied on atmospheric chemistry models, which showed similar surface temperature change patterns as found in the reanalysis data (e.g., Rozanov et al., 2005; Baumgaertner et al., 2011; Arsenovic et al., 2016). They reported a small cooling in the polar winter stratosphere due to EPP. However, the radiative effect of a polar night ozone loss should lead to a warming, which can also be found in reanalysis data (Lu et al., 2008; Seppälä et al., 2013). The simulated stratospheric cooling is attributed to a dynamical, adiabatic cooling caused by a decrease in the mean meridional circulation (Schoeberl and Strobel, 1978; Christiansen et al., 1997). Langematz et al. (2003) suggested that the weaker mean meridional circulation is caused by a decrease in midlatitude tropospheric wave forcing. The aforementioned model studies analyzing the climate impact of EPP relied on relatively few simulation years and applied complex forcings. Instead of prescribing ozone, these studies simulated EPP effects by changing the production of $NO_x$ and $HO_x$ and modelling the effects on ozone interactively. This could potentially be more realistic than simulations with prescribed ozone anomalies but introduces uncertainties related to the representation of chemistry and transport in the model, and renders the understanding of the effects more complicated as the ozone forcing varies in space and time. To avoid these difficulties and to obtain a clear signal-to-noise ratio, we use an idealized ozone forcing and a long simulation period.

Commonly, the effects of EPP are classified into direct and indirect effects (Randall et al., 2006, 2007). Direct effects are the effects of the local production of $NO_x$ and $HO_x$, whereas indirect effects are the effects of the $NO_x$ transport from the thermosphere to the stratosphere. Whereas most of the above mentioned studies discuss a mainly stratospheric ozone loss due to the indirect EPP effect, Andersson et al. (2014) suggested a potential climate influence of a mesospheric ozone loss due to the direct EPP effect. By using satellite observations they showed that $HO_x$ causes long-term variability in mesospheric ozone up to 34 % between EPP maximum and EPP minimum. Arsenovic et al. (2016) were the first to include the direct effect of $HO_x$ local production due to EPP in a chemistry-climate model. They found a similar mesospheric ozone loss as Andersson et al. (2014) and ultimately, reported a cooling over Greenland and a warming over Eurasia. However, Arsenovic et al. (2016) also considered the indirect effect of the $NO_x$ descent. Hence, the sole impact of a mesospheric ozone loss due to the direct EPP effect as suggested by Andersson et al. (2014) remains unclear.

This paper studies the circulation and climate impact of idealized mesospheric and stratospheric ozone losses that could be attributed to energetic particle precipitation. We use simulations with the Max Planck Institute Earth System Model (MPI-ESM) applying an idealized ozone forcing in either the mesosphere or the stratosphere. The idealized mesospheric ozone loss that we prescribe may be considered to be mostly a direct EPP effect, whereas the prescribed stratospheric ozone loss should be considered indirect. Additionally, we use a radiative transfer model to quantify the radiative forcing of ozone at different altitudes and months. Ultimately, we discuss whether an ozone loss in the middle atmosphere due to EPP has the potential to significantly alter the surface climate. Section 2 describes the MPI-ESM as well as the radiative transfer model. Section 3 links mesospheric and stratospheric ozone losses to changes in the atmospheric temperatures and winds. Finally, Section 4 summarizes and discusses the main outcomes and limitations of this study.

## 2   Models and numerical experiments

### 2.1   MPI-ESM: The Max Planck Institute Earth System Model

The Max Planck Institute Earth System Model (MPI-ESM; Giorgetta et al. (2013)) consists of the coupled atmospheric and ocean general circulation models, ECHAM6 (Stevens et al., 2013) and MPIOM (Jungclaus et al., 2013) as well as of the land and vegetation model JSBACH (Reick et al., 2013) and of the model for marine bio-geochemistry HAMOCC (Ilyina et al., 2013). We use the 'mixed-resolution' configuration of the model (MPI-ESM-MR). The ocean model uses a tripolar quasi-isotropic grid with a nominal resolution of 0.4° and 40 vertical layers. ECHAM6 is run with a triangular truncation at wave number 63 (T63), which corresponds to 1.9° in latitude and longitude. The vertical grid contains 95 hybrid sigma-pressure levels resolving the atmosphere from the surface up to 0.01 hPa. The vertical resolution is nearly constant (700 m) from the upper troposphere to the middle stratosphere and less than 1000 m at the stratopause. The time steps in the atmosphere and ocean are 450 and 3600 s, respectively.

The model has been used for many simulations within the CMIP5 (Coupled Model Intercomparison Project Phase 5) framework (Taylor et al., 2012). An overview of the dynamics of the middle atmosphere in these simulations is given by Schmidt et al. (2013). In this study, the preindustrial CMIP5 simulation (piControl) is used as reference. The forcing is con-

stant in time and uses pre-industrial conditions (1850 AD) for the greenhouse gases. Solar irradiance and ozone concentrations are averaged over a solar cycle (1844 – 1856 for the solar irradiance and 1850 – 1860 for ozone concentrations). No volcanic forcing is applied. A period of 150 years of this simulation is used.

In order to analyze the impact of ozone changes on the model climate, two additional experiments with reduced ozone concentration are carried out. In one experiment, the mesospheric ozone is reduced by 40 % between 0.01 hPa and 0.1 hPa polewards of 60° N (this is called "meso-$O_3$"). In the other experiment, stratospheric ozone is reduced by 20 % between 1 hPa and 10 hPa polewards of 60° N (this is called "strato-$O_3$"). We perform on-off experiments, whereas in reality EPP causes a constant (but variable) ozone loss. However, the magnitude of the prescribed ozone losses is based on satellite observations for winter conditions between years with high geomagnetic activity and years with low geomagnetic activity. In general, the impact of energetic particles is sporadic in the mesosphere, Andersson et al. (2014), however, showed that the direct $HO_x$ effect induces a long-term variability in mesospheric ozone up to 34 % from November to February in satellite data. Fytterer et al. (2015) and Damiani et al. (2016) revealed an upper stratospheric ozone loss between 10 – 15 % due to energetic particles for the Antarctic high latitudes in long-term measurements. Note that the applied ozone losses are slightly larger than the EPP influence diagnosed from observations. We use the stronger forcing to obtain a clear signal-to-noise ratio. However, this implies a potentially overestimated climate response.

To facilitate the experiment design, we applied the ozone losses constant over time. Although we concentrate our analysis on boreal winter high latitudes, this allows us to gain insights on boreal spring (i.e., the transition time from polar night to polar day). Observed ozone losses in summer are in general smaller than during winter, but this idealized setting allows an easy comparison of potential effects during the different seasons. In order to test whether in this experiment design the winter response is influenced through a preconditioning we repeated the experiments with ozone losses prescribed only from December to March. However, as the results are qualitatively very similar and differ only in the magnitude of the responses we discuss only the results of the experiments with ozone losses prescribed all year. Both experiments, with mesospheric and stratospheric ozone loss, are forced by the same conditions as the piControl experiment. Moreover, the simulations are restarted from the same year in the piControl experiment. This ensures that the ocean state is similar in all experiments. For both simulations 150 years are simulated.

The simplistic nature of our experiments is intended and, we think, useful. We chose this idealized experimental design in order to separate the climate impact of stratospheric and mesospheric ozone loss due to EPP and to identify the relevant mechanism how EPP affects the climate. Prescribing complex ozone reductions that vary in space, interseasonally and inter-annually, or simulating the ozone reduction interactively, might enable more realism but do not facilitate the identification of potential mechanisms. However, due to the simplification we cannot consider all features associated with EPP. In particular, three main effects are not taken into account: a) energetic particles enter the atmosphere only over the auroral oval regions (Hendrickx et al., 2015; Fytterer et al., 2015); b) the negative ozone signal due to EPP propagates from the stratopause in mid-winter to the lower stratosphere in spring within the polar vortex (Funke et al., 2017; Damiani et al., 2016); and c) the polar vortex can be shifted off the pole to regions with more solar radiation. We, instead, apply a constant ozone reduction between

the stratopause and mid-stratosphere (1 – 10 hPa) over the whole polar cap. The climate response in our simulations is likely overestimated as we reduce ozone over a larger latitudinal and altitude region than observations suggest.

In the Sections 3.2 and 3.3 the differences between the experiments and the control simulation (i.e., piControl) are analyzed. Statistical significance is calculated using the 95% confidence intervals assuming normally-distributed regression errors and using the 0.975 and 0.025 percentile of Student's t-distribution with the appropriate degrees of freedom. Properties of two simulations are considered statistically significantly different if the mean value of the control simulation is outside 95% confidence interval of the experiment.

## 2.2   The radiative transfer model PSrad

The radiative transfer scheme of MPI-ESM is based on the rapid radiation transfer suite of models optimized for general circulation models (RRTMG; (Mlawer et al., 1997; Iacono et al., 2008)). The RRTMG is widely used and its ability to calculate radiative forcing has been evaluated by Iacono et al. (2008). In its stand-alone version, it is used here to study the impact of ozone on heating rates. It is divided into sixteen bands in the longwave (1000 – 3 μm) and fourteen bands in the shortwave (12195 – 200 nm) (Clough et al., 2005). The spectral bands are chosen to include the major absorption bands of active gases. The major ozone absorption bands – Hartley band (200-310 nm), Huggins bands (310-350 nm), and Chappuis bands (410-750 nm) – are considered. However, absorption of oxygen at shorter wavelengths than 200 nm is missing, which could lead to an underestimation of the total heating rate in the mesosphere. The radiative transfer scheme is further described in Pincus and Stevens (2013) and Stevens et al. (2013) and onwards we will refer to it as the radiative transfer model "PSrad".

The shortwave and longwave components are calculated separately. Furthermore, optical properties for gases, clouds and aerosols are computed separately for longwave and shortwave and, finally, combined to compute the total heating rates. PSrad expects profiles of gases ($H_2O$, $N_2O$, $CH_4$, $CO$, $O_3$), profile of cloud parameters as well as additional parameters (e.g., albedo and zenith angle) as input. Additionally, $CO_2$ and $O_2$ are set to fixed values invariant with height. For all other gases, we use multi-year monthly means representative for the late 20[th] century provided by the atmospheric and chemistry model HAMMO-NIA (Hamburg Model for Neutral and Ionized Atmosphere; Schmidt et al. (2006)). For the albedo and cloud properties (e.g., cloud fraction, cloud water/ice content), multi-year monthly means from the piControl experiment are used. All quantities are extracted for 75° N. The zenith angle is calculated for 12 UTC at 75° N/0° E for the 15th of each month. The latitude of 75° N is chosen exemplary for a polar latitude. The results are insensitive to the actual latitude, the main difference at other polar latitudes is the length of the polar night. Note that the length of the polar night for an air pocket depends also on the altitude and on atmospheric dynamics (e.g., movement of the polar vortex). Both effects are omitted in this study. In our simulations we reduce ozone not depending on actual dynamics but over the whole polar cap (60 – 90 °).

To quantify the impact of ozone on the heating rates, we perform multiple runs in which for each run the ozone concentration of a single layer is set to 0 once. Then we take the differences between a control run and each single run. The differences of each run are, finally, added up for the estimation of the total heating rate. This method allows us to consider that layers of reduced ozone will lead to increased absorption of shortwave radiation in the layers directly below.

## 3 Results

### 3.1 Ozone effects on the heating rates

An ozone loss directly alters the atmospheric energy transfer. Before analyzing circulation and climate impacts due to ozone losses, we study the heating rate response using the radiative transfer model PSrad. The heating rates are calculated for the
polar latitude of 75° N (see Figure 1). As the effect of EPP is most important at the winter polar cap, we will concentrate our analysis on boreal winter high latitudes.

In the shortwave part of the spectrum, ozone strongly absorbs solar radiation and heats the whole atmosphere. The strongest heating (about 12 K/day) occurs in the uppermost stratosphere around 1 hPa. An ozone loss would, hence, result in a relative cooling due to reduced heating. The ozone heating and, hence, the cooling caused by an ozone reduction are getting smaller
for larger zenith angles and vanish in polar night.

In the longwave part of the spectrum, the radiative effect of ozone is highly temperature dependent. Ozone cools the atmosphere via infrared emission in the stratosphere and in warm regions of the mesosphere below 0.1 hPa (see Figure 1b). The strongest cooling (about -2 K/day) occurs at the stratopause. In the troposphere and in the cold regions of the mesosphere above 0.1 hPa, the absorption of outgoing radiation exceeds the infrared emission resulting in a heating of the atmosphere due
to ozone.

In total, the shortwave heating dominates all sunlit months. During polar night, ozone cools the atmosphere between 0.1 and 100 hPa and, hence, an ozone loss in the stratosphere and lower mesosphere results in a warming. Near the terminator (e.g., at 75° N in November and February), the net influence of ozone is more complex: At some altitudes ozone heats and at some it cools the atmosphere. The net radiative forcing of an ozone loss depends on when and where ozone is reduced. For example,
in November, a stratospheric (1 hPa) ozone loss leads to a heating, but a mesospheric (0.1 hPa) ozone loss to a cooling.

These results are in line with previous work. It is widely accepted that an ozone loss in spring and summer leads to a stratospheric cooling (e.g., Shine, 1986; Randel and Wu, 1999). Some studies analyzed the radiative forcing of a winter stratospheric ozone loss. Graf et al. (1998) showed that the observed stratospheric ozone loss in the late 20[th] century led to a winter warming and a summer cooling in a GCM. Using a radiative transfer model with fixed dynamical heating, Langematz et al. (2003) con-
firmed that a stratospheric ozone loss over the winter pole results in a small stratospheric radiative warming and a dominating stratospheric dynamical cooling. Shine (1986) showed that the shortwave cooling of the stratosphere due to an ozone loss dominates in all sunlit months the infrared heating due to an ozone loss. Recently, Sinnhuber et al. (2017) simulated a warming in mid-winter and a cooling in late winter and spring in the upper stratosphere for ozone losses explicitly induced by EPP.

The above stated results are confirmed by the actual heating rate anomalies induced by the applied ozone losses in the
experiments 'meso-$O_3$' and 'strato-$O_3$' (not shown). The heating rates are calculated at the first time step of the model at which the radiation is updated (1 January) excluding any feedbacks occurring only at later time steps. Note that the exact values may change for other months, especially depending on the sunlit area. Compared to the total heating rates of piControl, the changes in heating rates caused by a 40 % reduction of mesospheric ozone in polar night is very small (on average about -0.01 K/day and -0.4 %) and of a 20 % reduction of stratospheric ozone on average about 0.12 K/day and 2.6 % . This agrees with the estimate

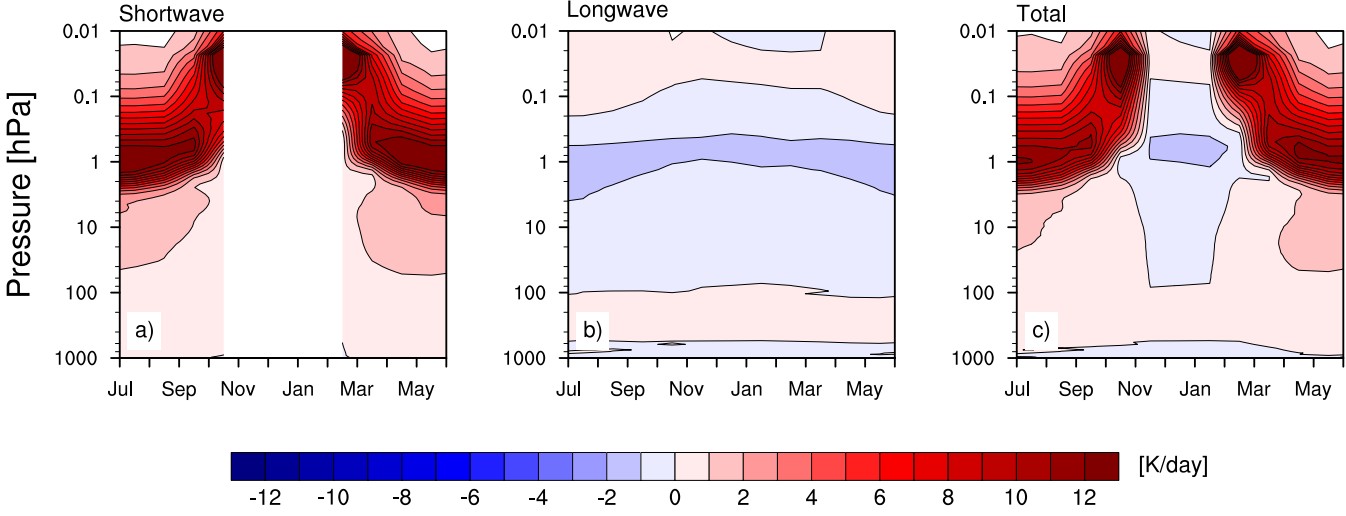

**Figure 1.** Monthly mean heating rates of ozone [K/day] for 75° N calculated by the radiative transfer model PSrad for (a) shortwave, (b) longwave and (c) total (shortwave + longwave) radiation.

of Sinnhuber et al. (2017), who simulated a change of 0.1 K/day in the winter stratospheric heating rate due particle-induced ozone loss. The change in heating rates due to the stratospheric ozone change is in the range of solar UV forcing (0.1 K/day) (Gray et al., 2010).

## 3.2 Climate effects of a mesospheric ozone loss

5   As changes in heating rates due to a reduced ozone during polar night are small, one might reason that climate impact of a winter polar ozone loss is small. But large effects may occur in regions slightly outside the polar night. Furthermore, several studies suggested that changes in the heating rates due to a winter polar ozone loss leads to a dynamical cooling (e.g., Langematz et al., 2003; Baumgaertner et al., 2011; Arsenovic et al., 2016), whereas the initial radiative forcing suggests a warming. Hence, we further analyze the climate impact of a winter polar ozone loss. As large variations in the polar vortex can propagate downward
10  and affect the surface climate, we first concentrate on the circulation changes of the middle atmosphere due to an ozone loss, which are a prerequisite for a potential climate impact of EPP. In the following, we analyze the climate effect of an idealized polar mesospheric ozone loss, while in Section 3.3 we analyze the climate effect of an idealized polar stratospheric ozone loss.

Figures 2a and 2d show the zonal mean temperature and zonal wind simulated for boreal winter (December - February). Main observed characteristics of the zonal mean temperature, e.g., the stratopause tilt from the summer towards the winter
15  pole, are well reproduced. The changes in the zonal mean zonal wind are consistent with the temperature changes via the thermal wind balance. In most regions, the difference between meso-O$_3$ and piControl is very small (see Figures 2b and 2e).

Near the winter pole, a dipole structure emerges with cooling in the upper stratosphere and warming in the mesosphere. According to our radiative transfer calculations a mesospheric winter polar ozone loss should lead to a cooling. However, the

temperature differences are small (below 1 K) and not significant at the 95 % level. As the applied forcing is very small, small and low significant values are expected. At the winter pole, the polar vortex slightly weakens, whereas the mesospheric winds strengthen: these differences are not significant. The signal is only slightly stronger but still insignificant if winters with major sudden stratospheric warmings (SSW) are excluded (not shown). As stated above, large variations in the winter polar vortex can propagate to the surface influencing the surface climate. However, the changes reported here are small. The anomalies reaching the troposphere are statistical artifacts. Indeed, the surface temperature reveals no statistically significant change (not shown).

Although, the temperature and wind signals are not statistically significant after 150 simulated years, nevertheless, it makes sense to analyze if the signals could have a physical explanation and not be purely accidental. Note that with fewer simulation years apparently very different results can be obtained. Analyzing different simulation periods we obtain mesospheric warming and cooling of apparent significance. Particularly, we calculated a statistically significant weakening of the polar vortex when using only the first 80 simulation years. We can not identify a model drift in the experiments, which could explain the disagreement between the 150-year and 80-year runs. However, the model simulates variability on time-scales up to multi-decadal, which is common in many climate models (Sutton et al., 2015), and might cause the apparently different responses to ozone reduction in different sub-periods of the 150-year simulation. The high degree of internal variability of the winter polar stratosphere can obviously create wrong apparent signals. The most dramatic demonstration of this variability are major sudden stratospheric warmings (SSW), which occur on average about 6 times per decade in the Northern Hemisphere (see Charlton and Polvani (2007) for more information on SSW). A short simulation period may lead to an over-representation or under-representation of SSWs. Over our whole simulation period (150 years) the number of major SSWs is balanced in all three experiments. In total, there are 102 events in piControl, 99 events in meso-$O_3$ and 109 events in strato-$O_3$ (using a reversal of the zonal wind at $60°$ N and 10 hPa as criterion of a major SSW occurrence).

### 3.3 Climate effects of a stratospheric ozone loss

In this section, we analyze the climate effect of an idealized polar stratospheric ozone loss. Figures 2a and 2d show the zonal mean temperature and zonal wind simulated for boreal winter (December - February) for piControl, and Figures 2c and 2f the difference between strato-$O_3$ and piControl. The winter stratosphere warms due to an ozone loss as expected from the calculations with the radiative transfer model. As a consequence of the warming, the stratospheric winds weaken. The small mesospheric cooling likely results from enhanced eastward momentum deposition from gravity waves as shown by Lossow et al. (2012). Our results are in line with earlier studies. Seppälä et al. (2013) and Lu et al. (2008) identified a warming in the polar winter upper stratosphere due to EPP in reanalysis data, but their magnitude is much stronger (5 K) than in our simulations. Regarding the zonal wind response, the two studies differ from each other. Seppälä et al. (2013) analyzed a strengthening of the polar vortex with enhanced equatorward planetary waves, whereas Lu et al. (2008) analyzed a weakening of the polar vortex. The statistically significant warming of the summer mesopause is an indication of inter-hemispheric coupling as discussed by Karlsson and Becker (2016) and also persists for winters without a sudden stratospheric warming event.

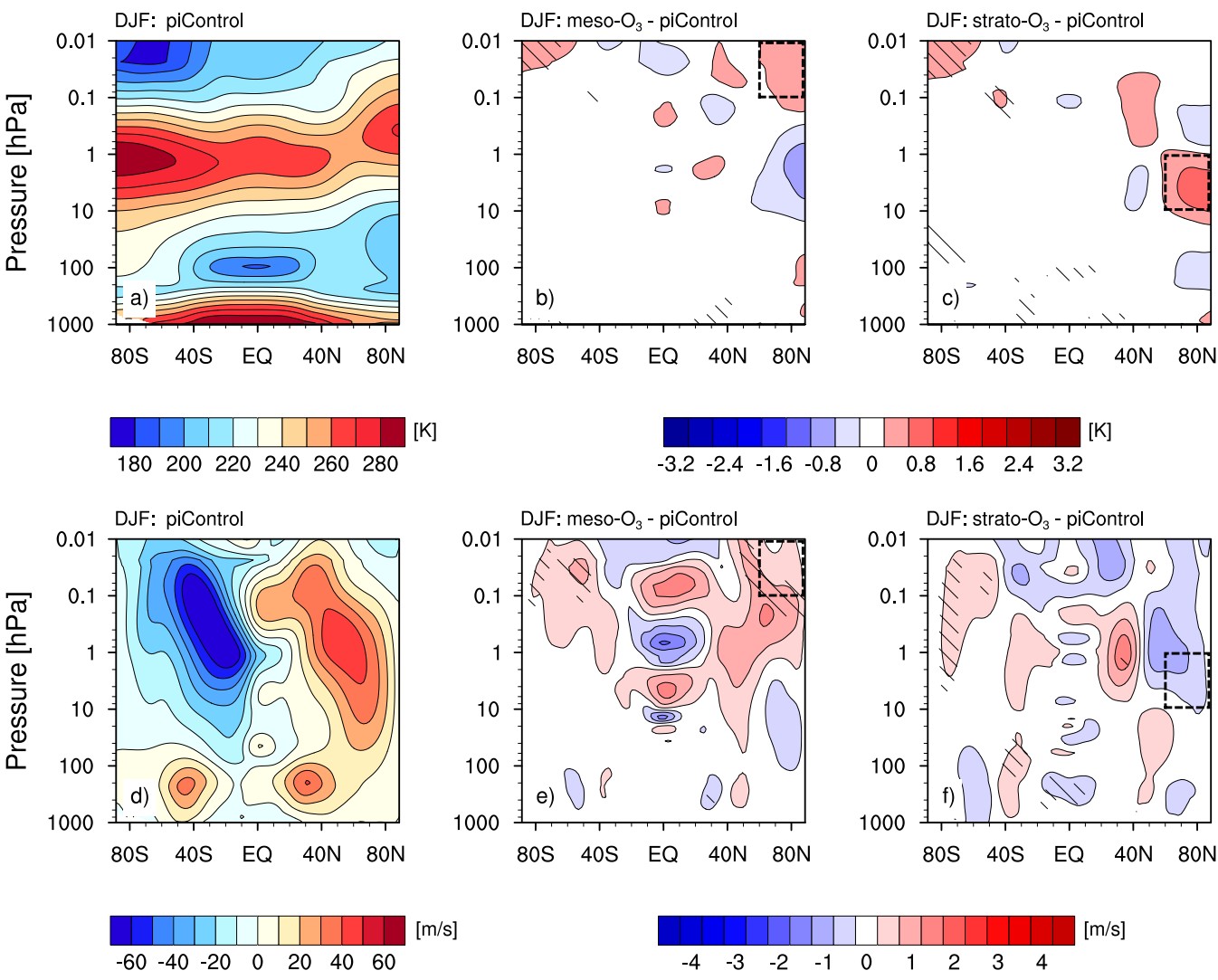

**Figure 2.** (upper row) Zonal mean temperature [K] and (lower row) zonal mean zonal wind [m/s] averaged over December - February (DJF) for (a,d) piControl, (b,e) the difference between meso-O$_3$ and piControl and (c,f) the difference between strato-O$_3$ and piControl. Shaded areas are statistically significant at the 95 % confidence interval. The black, dashed boxes highlight the regions where ozone is reduced.

Figure 2 shows only changes for the mean over December to February, while the radiative transfer model suggests that the month-to-month variability of the forcing is large. To study whether the impact of a stratospheric ozone loss differs over the course of the winter, we analyze the monthly means of temperature and zonal wind (see Figure 3). An ozone loss during most of the polar night (except December) leads to a warming, whereas at all other times and locations it leads to a cooling. This agrees with the calculations of the radiative transfer model and with our assumption that the winter cooling is not affected by a strong summer warming. However, the cooling in December is unexpected from the radiative transfer modeling. Kodera and Kuroda (2002) argued that the polar winter atmosphere transits from a radiatively controlled state in early winter to a dynamically controlled state in late winter. Given the opposite sign of the diabatic forcing, the simulated cooling must be dynamically caused already in December. This is in agreement with early model studies which showed that uniform ozone losses lead to dynamical cooling at the boreal winter polar latitudes (e.g., Schoeberl and Strobel, 1978; Kiehl and Boville, 1988). Langematz et al. (2003) suggested that the dynamical cooling is due to a weakening of the mean meridional circulation related to reduced wave forcing caused by a reduction of mid-latitude wave flux into the stratosphere. Similarly, in our simulations we find a (albeit not significant) reduction of the zonal mean eddy heat flux at 100 hPa in the midlatitudes from December to March (not shown). This may be caused by enhanced wave reflection as suggested by Lu et al. (2017) for the dynamical response to 11-year solar irradiance forcing. The dynamically induced cooling in December also occurs in simulations in which the ozone is only reduced from December to March (not shown). Also Baumgaertner et al. (2011) reported a dynamical cooling in the winter polar stratosphere due to EPP. However, in their model the cooling dominates the winter (DJF) signal, whereas we obtain a small warming for the DJF average (see Figure 2). The magnitude of the signal decreases in our simulations, especially in late winter, if we exclude all seasons with a SSW (not shown).

The zonal wind changes consistently with the temperature changes via the thermal wind balance. Simultaneously with the warming (cooling), the polar wind weakens (strengthens). Anomalies in the polar vortex occasionally reach the troposphere (e.g., the strengthening in November or the weakening in December or February). Although, most of those changes are not significant, some disturbances in the polar vortex may still force the surface temperature (see Figure 4). In our simulations for boreal winter, stratospheric ozone loss cools large parts of the northern high latitudes from northern Europe to Eurasia and over northern America. Excluding all winter with a SSW strengthens the cooling over northern America (not shown). Over Greenland and the pole, the surface warms. This is consistent with the weakening of the zonal wind in December (see Figure 3 d). However, most changes are small and not significant. Seppälä et al. (2009) and Baumgaertner et al. (2011) analyzed statistically significant changes in surface temperature: A warming over Eurasia of about 1.5 K and a cooling over northern America of about -1 K. Compared to both studies the amplitude of our signal is much smaller. The weaker signal also persists if we exclude all winters with a SSW (not shown). However, Baumgaertner et al. (2011) based their study on only nine simulated years and we have shown that the large variability in the polar winter stratosphere can cause wrong apparent signals if the ensemble is not large enough. Note that another difference between our and their study is that our model is coupled to an interactive ocean. Seppälä et al. (2009) could not rule out that their results are by chance induced by the Northern Annual Mode.

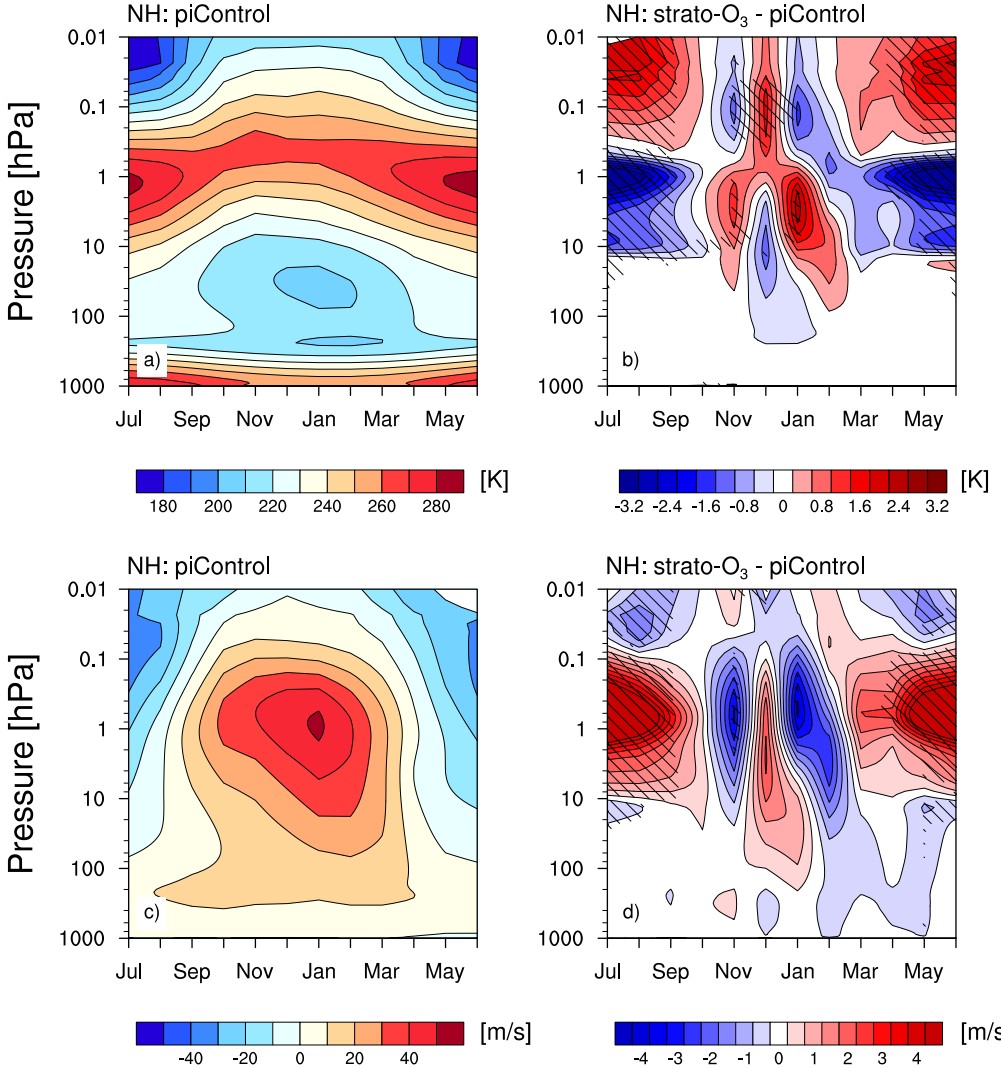

**Figure 3.** Monthly mean (upper row) temperature averaged between $60°$ N and $90°$ N [K] and (lower row) zonal wind [m/s] for $60°$ N for (a,c) piControl and (b,d) the difference between strato-$O_3$ and piControl. Shaded areas are statistically significant at the 95 % confidence interval.

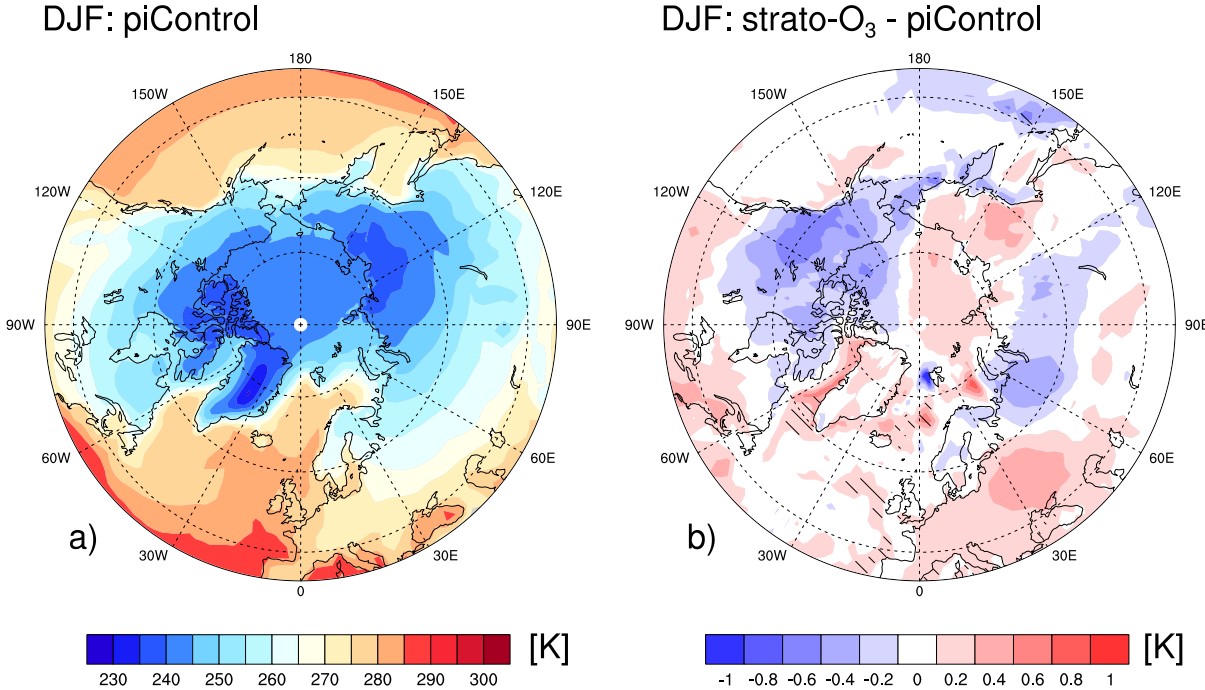

**Figure 4.** Surface temperature [K] averaged over December - February (DJF) for the Northern Hemisphere for (a) piControl and (b) the difference between strato-O$_3$ and piControl. Shaded areas are statistically significant at the 95 % confidence interval.

## 4   Summary and conclusion

In this study, we analyzed the climate impact of idealized mesospheric and stratospheric ozone losses. Although this study is motivated from the enhancement of NO$_x$ due to energetic particle precipitation (EPP), the results presented here could also be applied to other processes causing ozone destruction. We lie the focus on boreal winter. The radiative forcing of polar ozone is calculated by the radiative transfer model PSrad. In sensitivity studies with the Max Planck Institute Earth System Model (MPI-ESM), we applied idealized ozone losses either of 40 % in the winter polar mesosphere or of 20 % in the winter polar stratosphere. This simplified design facilitates the identification of the processes relevant for possible climate responses.

Recently, Andersson et al. (2014) showed that the direct EPP-HO$_x$ effect induces large long-term variability in winter mesospheric ozone. They suggested that these large changes may have an impact on climate. Following their idea, we analyzed the atmospheric response to a mesospheric ozone loss. We found that the winter atmospheric changes due to a mesospheric ozone loss in our model are negligible. Calculations with a radiative transfer model showed that the radiative forcing of mesospheric ozone is very small during polar night, which makes the small dynamic response plausible.

Several studies analyzed the climate effect of a stratospheric ozone loss due to EPP. Seppälä et al. (2009) calculated a correlation of the winter surface temperature and energetic particle precipitation in reanalysis data. However, they could not rule out an accidental occurrence of the correlation. Since then several model studies tried to establish a physical link between EPP

and climate (Baumgaertner et al., 2011; Rozanov et al., 2012; Arsenovic et al., 2016). In all these model studies, a dynamical cooling of the winter polar stratosphere due to energetic particle precipitation was simulated. In our model, a stratospheric ozone loss during polar night (except December) results in a warming, whereas at all other times and locations it leads to a cooling. This agrees with the calculations of the radiative transfer model. We obtained a cooling during December due to stratospheric ozone loss caused by a reduced vertical wind. However, the changes in the polar winter stratosphere are small and not significant in our model. Consequently, also the impact on the simulated winter surface temperature is weak. In contrast to the above mentioned studies, in our experiment the dynamical feedback leading to the stratospheric cooling is not dominant throughout the boreal winter. This is also true if we restrict the ozone loss to December to March. However, the earlier model studies were based on only a few simulation years. Using only the first 80 years of our simulations we obtained false positives. The high degree of internal variability of the polar vortex can create wrong apparent signals.

As the radiative forcing of our prescribed mesospheric ozone loss is negligible, a significant climate impact of a mesospheric ozone change as suggested by Andersson et al. (2014) seems unlikely. Our experimental design would likely rather overestimate the climate impact of EPP than underestimate it. However, our simulations indicate only small changes in the stratospheric circulation and temperature and a weak impact on surface temperature. We encourage more research on the effects of EPP as the climate impact of stratospheric ozone losses due to EPP is not as clear as often thought and the underlying processes are not well understood. The upcoming CMIP6 model intercomparison may help to resolve those open points, because energetic particle forcing is recommended - for the first time - as part of the solar forcing. (Matthes et al., 2017). Especially the role of wave reflection for the coupling mechanism between stratosphere and troposphere needs to be clarified. Furthermore, the catalytic destruction of ozone by $NO_x$ works only effectively if sunlight is available. The influence of EPP induced $NO_x$ may be larger near the terminator.

Moreover, the simplified experimental design has its limits. It is suitable to address the different processes related to direct and indirect EPP impacts and the identification of mechanisms for possible climate responses. However, we cannot rule out that the time and altitude dependence of the ozone loss caused by the downward transport of ozone and nitrogen oxides in the polar vortex is important. But we obtain qualitatively very similar results if the ozone is only reduced from December to March.

Finally, although previous studies have shown that MPI-ESM reproduces stratospheric temperature responses to forcings reasonably well (e.g., Bittner et al., 2016; Schmidt et al., 2013), the possibility remains that the model's sensitivity to ozone loss is biased low. To address this, we would like to encourage multi-model studies on EPP climate impact as currently suggested for the third phase of the SOLARIS-HEPPA project, which investigates solar influences on climate as part of the 'Stratosphere-troposphere Processes And their Role in Climate' (SPARC) project.

## 5  Code and data availability

Primary data and scripts used in the analysis and other supplementary information that may be useful in reproducing the author's work are archived by the Max Planck Institute for Meteorology and can be obtained by contacting publications@mpimet.mpg.de.

*Acknowledgements.* The authors acknowledge scientific and practical input from Matthias Bittner and Elisa Manzini. This study was supported by the Max-Planck-Gesellschaft (MPG) and computational resources were made available by Deutsches Klimarechenzentrum (DKRZ) through support from Bundesministerium für Bildung und Forschung (BMBF). The authors thank two anonymous referees for useful comments and suggestions.

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
