# Peer review of "Climate Impact of Idealized Winter Polar Mesospheric and Stratospheric Ozone Losses as caused by Energetic Particle Precipitation"

_Atmospheric Chemistry and Physics, 2017_

## Referee Comment (RC1) · Anonymous Referee #1 · 26 Jul 2017

In this paper, simplified model experiments are carried out to investigate the impact of ozone loss induced by energetic particle precipitation on atmospheric temperatures and dynamics from the mesosphere down to the surface. The topic is highly relevant at the moment, as energetic particle precipitation is recommended as part of the solar forcing for the upcoming CMIP-6 model experiments (Matthes et al., ACP, 2017). The results therefore are of great interest, and the paper is also very clearly structured and well written. However, there are three points which need to be addressed before the paper can be published in ACP: a) the setup of the model experiments does not reflect the temporal and spatial structure of the direct and indirect particle impact as it is known from observations; b) some observation of the temperature response of

the winter-time stratosphere to geomagnetic activity exist (e.g., Lu et al., JGR, 2008; Seppaelae et al., JGR, 2013) but are not used here to compare the results of this model run (actually the observed amplitude is much larger than the results shown here). This comparison needs to be included as it provides ground truth to estimate how realistic the modeled response of the troposphere is; c) the estimation of significance using a t-test is not applicable to the high-latitude Northern hemisphere winter, where due to the occurence of strong sudden stratospheric warmings the underlying distribution is bimodal.

These as well as a few more minor points are discussed in more detail below.

Page 1, lines 11 to page 2, line 8: the impact of energetic particle precipitation on the middle and lower atmosphere has been investigated since the 1970th, and a lot more has been published than referenced here. In particular there are two recent review papers which summarize the state of the art (Sinnhuber et al., Sur Geo, 2012; Mironova et al., Space Sci Rev, 2015), as well as reports on observations of a) the temporal and spatial structure of the indirect effect in different trace species (e.g., Hendrickx et al., JGR, 2015; Fytterer et al., JGR, 2015; Sinnhuber et al., JGR, 2016; Friederich et al., ACP, 2014); b) the temporal and spatial structure of the indirect effect in NOy (e.g., Funke et al., JGR, 2014a, b) and ozone (e.g., Fytterer et al., ACP, 2015; Damiani et al., GRL, 2016; Kazutoshi et al., ACP, 2017), c) the impact of the indirect effect on stratospheric temperatures and winds in the Northern hemisphere winter and spring (e.g., Lu et al., JGR, 2008; Seppaelae et al., JGR, 2013), and d) the response of tropospheric weather patterns to geomagnetic activity (e.g., Seppaelae et al., JGR, 2009; Maliniemi et al., JGR, 2014). Observations provide the ground truth your model study has to compare to, so should be summarized here.

Page 3, lines 22-25, description of model experiments with reduced ozone loss: the scenarios differ quite substantially from what is known about particle induced ozone loss from observations of the direct and indirect impact. They are very much simplified, and of course there is justification for carrying out very simple model studies. However, you should be aware how they differ from reality (as provided by observations), and discuss this carefully. Direct impact, mesospheric ozone: the direct impact has been shown to occur in sporadic events which are mostly short-lived (one day to a few days), but can occur in a periodicity related to solar rotation (27 days, 13.5 days, 18 or 9 days). It is restricted clearly to geomagnetic latitudes corresponding to the auroral oval (about 60-75° geomagnetic latitude). Implying this impact onto the whole polar cap should lead to an overestimation of this impact (see e.g., Hendrickx et al., JGR, 2015; Fytterer et al., JGR, 2015; Sinnhuber et al., JGR, 2016; Friederich et al., ACP, 2014). The indirect effect has been observed in every winter where observations in polar night have been available (Funke et al., 2014a,b). The impact of ozone is characterized by a downwelling negative anomaly starting in the upper stratosphere in mid-winter, and moving downwards to below 30 km in spring; it is restricted to the polar vortex (e.g., Fytterer et al., ACP, 2015; Damiani et al., GRL, 2016; Kazutoshi et al., ACP, 2017). Amplitudes are generally less than 20%, however it should be pointed out that observations show the difference of years with high to years with low geomagnetic activity; as the indirect effect occurs in every winter, see above, this is something different to the model experiments, which compare years with high activity to years with no activity, something that in reality doesn't happen even during deep solar minimum.

Page 4, lines 5-9, determination of statistical significance: using a t-test implies a distribution of temperatures which is random around a mean state. However, in the Northern hemisphere polar winter, this is obviously not the case: years with sudden stratospheric warmings are not outliers of the mean atmospheric state distribution, they belong to a different distribution: the distribution of temperatures do not approach a normal distribution (as student's t-distribution), but is bimodal, with one mode for the years without, and one mode for the years with warmings. Therefore, you can only use the t-test separately for years with and without warmings (if the distribution of those years is indeed symmetric, which maybe you should check before doing a statistical test); it is definitely not applicable, and therefore meaningless, for the whole sample of winters with and without warmings.

Page 4, line 25-27: I eventually understood what you did there, but the sentence was difficult to follow. Maybe you can clarify it.

Page 5, lines 12-13: there is one publication in ACPD at the moment which shows the same impact on heating rates (Sinnhuber et al., 2017) using a slightly different approach to yours. The results seem comparable, and I would encourage you to discuss/compare those results to yours.

Page 5, line 28: a change in the heating rate of 10% as for your stratospheric ozone experiment means a change of 0.1-0.2 K/day (see Figure 1). Observations and also the model study by Sinnhuber et al., ACPD, 2017, imply that this change in the stratosphere is not sporadic, but persists for several weeks, implying a warming during mid-winter of a few K. That is actually not a small change, and also in line with observations of the temperature response due to high geomagnetic activity in the high-latitude upper stratosphere (e.g., Lu et al., 2008; Seppaelae et al., 2013).

Page 5-8, discussion of statistical significance: a t-test is just not applicable if you combine years with and years without SSWs, see my comment above. I think you should study the change in years with and without warmings separatedly; then you can provide a robust measure of the significance. Also, this would make the results more comparable to the observations shown in Seppaelae et al., 2013, for the stratospheric response, as they also analyze years without warmings.

Page 8, line 4: the impact in the winter-time high latitude upper stratosphere temperatures you show in Figure 2 has a similar structure to observed temperature and wind field changes for years with high geomagnetic activity (Lu et al., 2008; Seppaelae et al., 2013). However, the amplitude of the warming is much smaller (about one order of magnitude?) than in the observations. This comparison to observations needs to be discussed here.

Page 8, line 8: the interhemispheric coupling is evident in both the meso-O3 and the strato-O3 experiments as a "statistically significant" change in the summertime upper

mesosphere. However, this is more likely an affect of SSWs?

Page 8, line 25-30: The patterns and amplitudes you observe here should be compared to observations (Seppaelae et al., 2009; Maliniemi et al., 2014). However, as the amplitudes of your stratospheric warming appears to be much lower than observed, I would expect the impact on the troposphere also to be low compared to observations. Another point: Seppaelae et al., 2009 show that the impact on surface temperatures is different, with larger amplitudes, when years with SSWs are not considered. You should separate years with and years without warmings here as well. Can you reproduce their result regarding the impact of warmings? Again, a t-test is not applicable if you use years with and without warmings.

Page 11, 11: "Our results suggest that the climate impact of an ozone loss due to EPP is small" considering that the impact of particle precipitation in your analysis is masked by the strong variability implied on the Northern hemisphere winter atmosphere by sudden stratospheric warmings, and your results of the stratospheric impact strongly underestimate the observed response of the stratosphere, you can not draw this conclusion at this point.

---

## Referee Comment (RC2) · Anonymous Referee #2 · 18 Aug 2017

General comments:

The manuscript presents the response of the atmosphere and surface temperature to the introduced permanent decrease of the ozone concentration in the mesosphere and upper stratosphere simulated with the MPI-ESM model. The forcing was designed to mimic the ozone depletion by hydrogen and nitrogen oxides formed by the precipitating energetic particles. The subject of the manuscript is appropriate for ACP because it addresses widely discussed during the last decade question about possible influence of the energetic particles on the atmosphere, ozone and surface air temperature. The manuscript is well written, the most of relevant publications are cited, the figures are

clear. However, the manuscript does not look mature because the bold conclusions cannot really be supported by the presented results. It seems obvious for the authors because in the summary they formulate why the results are not convincing and what to do to make them better. Therefore, I cannot recommend publication in the present form.

Main issues:

1. The experimental design is too simplified. It resembles the ozone loss due to EPP obtained from the observations and models however substantially differs in the time evolution and distribution in space. Application of realistic ozone depletion scenarios could lead to very different results. If the authors do not know the implications of the chosen scenario (as it is said in the summary) what potential readers could learn from the paper? There are several aspects of the problem such as shift of the vortex from the pole and intensified ozone influence on solar radiation heating or interaction of the propagating disturbance with internal variability modes like PJO. These effects are automatically taken into account in the models considered all relevant to EPP processes, but they are missed if too simplified approach is applied. The simplest way to avoid the problem is to eliminated connection with EPP. Actually, the introduced ozone depletion scenario in the upper stratosphere is closer to the influence of halogens.

2. I found interesting a large disagreement between the results of 80 and 150-year long runs. I guess, this phenomenon should be understood and explained with more details. I am not convinced that it is just the results of inter-annual variability. If so all modeling community is in a huge trouble. Did the authors check the presence of any model drift?

3. The authors frequently discuss not statistically significant responses. I have noticed that almost all results presented in Figure 2 and 4 are not significant. It is rather interesting why the applied model is not sensitive to 20% decrease of the ozone in the polar upper stratosphere. There were several publications (mentioned in the introduction) claiming significant response of the atmosphere to the observed ozone depletion in the last decades of 20th century and the ozone depletion scenario is close to what is used in the manuscript. Some discussion of this issue is necessary.

4. Section 3.1: The use of 75N should be better motivated if the authors would like to wire these results with ozone depletion due to EPP. If the ozone depletion occurs inside polar vortex then 75N is not representative because huge ozone influence on solar heating rate outside polar night area will dominate over very small longwave effect. It should be also considered that in the Northern hemisphere the vortex is not stable and tends to move from the pole out of the polar night area.

Minor issues:

1. Page 2, line 2: if –> of

2. Page 2, line 4: Langematz et al. (2003) showed tiny direct LW warming (Fig.7) , but the resulting stratosphere is cooler (Fig.8). Graf et al., (1998) showed the response in the lower stratosphere (70 hPa).

3. Page 3, line 23-25, line 31: The ozone depletion scenario is too simplified.

4. Section 2.2: The radiation code is not described. The references do not provide satisfactory information about the treatment of solar (e.g., spectral range coverage, spherical) and infrared (e.g., LTE treatment) radiation. The standard version of the RRTMG does not include wavelengths shorter 200 nm and therefore the heating rate in the mesosphere should be heavily underestimated due to the absence of Lyman-alpha line and Schumann-Runge bands. How it is treated in PSrad?

5. Page 4, lines 16-18: I do not understand what means "separately . . .and then combined". Why $CO_2$ is not in the input list. Is it not included in PSrad?

6. Page 4, line 24: Actually, the length of the polar night depends on the altitude and at 80 km it could well be shifted by one month relative to the surface. In Figure 1 this effect is absent, which affects the results in the mesosphere.

7. Page 5, line 4: The maximum of the ozone VMR is normally around 6 hPa for this location. What ozone profiles were used?

8. Page 5, line 33: I guess, Langematz et al. (2003) showed the same.

9. Page 6, line 9: 75N is not really representative (see above).

10. Page 8, line 5: 75N is not really representative (see above). This result disagrees with Langematz et al. (2003, see their Figure 7 and 8).

11. Page 8, line 15: statein –> state in

---

## Author Comment (AC1) · 13 Oct 2017

We thank the reviewer for the assessment of our work and the help to connect it better to previous studies. Below we reply point by point, first showing the reviewers comments in italic and blue followed by our response. To avoid confusion, we refer to graphics shown in this document as Fig. and graphics shown in the paper manuscript as Figures.

*In this paper, simplified model experiments are carried out to investigate the impact*

*of ozone loss induced by energetic particle precipitation on atmospheric temperatures and dynamics from the mesosphere down to the surface. The topic is highly relevant at the moment, as energetic particle precipitation is recommended as part of the solar forcing for the upcoming CMIP-6 model experiments (Matthes et al., ACP, 2017). The results therefore are of great interest, and the paper is also very clearly structured and well written. However, there are three points which need to be addressed before the paper can be published in ACP: a) the setup of the model experiments does not reflect the temporal and spatial structure of the direct and indirect particle impact as it is known from observations; b) some observation of the temperature response of the winter- ime stratosphere to geomagnetic activity exist (e.g., Lu et al., JGR, 2008; Seppaelae et al., JGR, 2013) but are not used here to compare the results of this model run (actually the observed amplitude is much larger than the results shown here). This comparison needs to be included as it provides ground truth to estimate how realistic the modeled response of the troposphere is; c) the estimation of significance using a t-test is not applicable to the high- latitude Northern hemisphere winter, where due to the occurence of strong sudden stratospheric warmings the underlying distribution is bimodal. These as well as a few more minor points are discussed in more detail below.*

*Page 1, lines 11 to page 2, line 8: the impact of energetic particle precipitation on the middle and lower atmosphere has been investigated since the 1970th, and a lot more has been published than referenced here. In particular there are two recent review papers which summarize the state of the art (Sinnhuber et al., Sur Geo, 2012; Mironova et al., Space Sci Rev, 2015), as well as reports on observations of a) the temporal and spatial structure of the indirect effect in different trace species (e.g., Hendrickx et al., JGR, 2015; Fytterer et al., JGR, 2015; Sinnhuber et al., JGR, 2016;Friederich et al., ACP, 2014); b) the temporal and spatial structure of the indirect effect in Noy (e.g., Funke et al., JGR, 2014a, b) and ozone (e.g., Fytterer et al., ACP, 2015; Damiani et*

[Figure]

*al., GRL, 2016; Kazutoshi et al., ACP, 2017), c) the impact of the indirect effect on stratospheric temperatures and winds in the Northern hemisphere winter and spring (e.g., Lu et al., JGR, 2008; Seppaelae et al., JGR, 2013), and d) the response of tropospheric weather patterns to geomagnetic activity (e.g., Seppaelae et al., JGR, 2009; Maliniemi et al., JGR, 2014). Observations provide the ground truth your model study has to compare to, so should be summarized here.*

We followed the suggestion of the reviewer and included a summary of the observational record on the ozone loss due to EPP as well as the impact on the stratospheric temperature and zonal wind to the introduction. We included a large number of the suggested references.

*Page 3, lines 22-25, description of model experiments with reduced ozone loss: the scenarios differ quite substantially from what is known about particle induced ozone loss from observations of the direct and indirect impact. They are very much simplified, and of course there is justification for carrying out very simple model studies. However, you should be aware how they differ from reality (as provided by observations), and discuss this carefully. Direct impact, mesospheric ozone: the direct impact has been shown to occur in sporadic events which are mostly short-lived (one day to a few days), but can occur in a periodicity related to solar rotation (27 days, 13.5 days, 18 or 9 days). It is restricted clearly to geomagnetic latitudes corresponding to the auroral oval (about 60-75 ° geomagnetic latitude). Implying this impact onto the whole polar cap should lead to an overestimation of this impact (see e.g., Hendrickx et al., JGR, 2015; Fytterer et al., JGR, 2015; Sinnhuber et al., JGR, 2016; Friederich et al., ACP, 2014). The indirect effect has been observed in every winter where observations in polar night have been available (Funke et al., 2014a,b). The impact of ozone is characterized by a downwelling negative anomaly starting in the upper stratosphere in mid-winter, and moving downwards to below 30 km in spring; it is restricted to the polar vortex (e.g.,*

*Fytterer et al., ACP, 2015; Damiani et al., GRL, 2016; Kazutoshi et al., ACP, 2017).
Amplitudes are generally less than 20%, however it should be pointed out that observations show the difference of years with high to years with low geomagnetic activity;
as the indirect effect occurs in every winter, see above, this is something different to
the model experiments, which compare years with high activity to years with no activity,
something that in reality doesn't happen even during deep solar minimum.*

We agree with the reviewer that our description of the experiments was too brief. We
added two paragraphs to Section 2.1, also taking into account the comment of Reviewer #2. We still believe that our experimental design is justified, because this allows
us a clear signal-to-noise ratio and long simulation periods in order to gain as much
insights in the processes governing the climate impact of EPP. But we now added a
discussion on how our experiments differ from the observational record. In particular,
the lack of a vertical propagation of the signal, the shift of the polar vortex and the
EPP estricted to the auroral oval are now discussed. We would also like to thank the
reviewer for providing such an extended list of references, from which several are now
cited in the manuscript.

*Page 4, lines 5-9, determination of statistical significance: using a t-test implies a distribution of temperatures which is random around a mean state. However, in the Northern
hemisphere polar winter, this is obviously not the case: years with sudden stratospheric
warmings are not outliers of the mean atmospheric state distribution, they belong to a
different distribution: the distribution of temperatures do not approach a normal distribution (as student's t-distribution), but is bimodal, with one mode for the years without,
and one mode for the years with warmings. Therefore, you can only use the t-test
separately for years with and without warmings (if the distribution of those years is indeed symmetric, which maybe you should check before doing a statistical test); it is
definitely not applicable, and therefore meaningless, for the whole sample of winters*

*with and without warmings.*

We analyzed the probability density functions (pdfs) for the variables shown in the paper. Fig. 1 shows exemplary the pdfs for winter polar mean (left) temperature averaged over (60 – 90N) and (right) zonal wind at 60N between 1 – 10 hPa. This corresponds to Figure 2 in the paper. Although the distribution is not smooth, it resembles more a normal distribution than a bi- modal distribution. We don't think that the notion of two distinct states, with and without an SSW is correct. There is a spectrum of major SSWs of very different peak intensities and durations which smoothly transitions into winter states without major SSWs which however often include minor SSW events of again different characteristics. This is also reflected by the fact that different SSW definitions may identify different cases. We think it is sufficiently justified to use Student's t-Test as it has been done in many earlier publications, e.g., by Seppälä et al. (2009), Arsenovic et al. (2016) and Gray et al. (2012). We decided to stick to the Student's t-test.

*Page 4, line 25-27: I eventually understood what you did there, but the sentence was difficult to follow. Maybe you can clarify it.* Done.

*Page 5, lines 12-13: there is one publication in ACPD at the moment which shows the same impact on heating rates (Sinnhuber et al., 2017) using a slightly different approach to yours. The results seem comparable, and I would encourage you to discuss/compare those results to yours. Thank you for pointing us to this paper.* We added a comparison to this paper in Section 3.1.

*Page 5, line 28: a change in the heating rate of 10% as for your stratospheric ozone experiment means a change of 0.1-0.2 K/day (see Figure 1). Observations and also the*

*model study by Sinnhuber et al., ACPD, 2017, imply that this change in the stratosphere is not sporadic, but persists for several weeks, implying a warming during mid-winter of a few K. That is actually not a small change, and also in line with observations of the temperature response due to high geomagnetic activity in the high-latitude upper stratosphere (e.g., Lu et al., 2008; Seppaelae et al., 2013).* We added the absolute values of the change in heating rates to the manuscript and compare it with the model study of Sinnhuber et al. (2017).

*Page 5-8, discussion of statistical significance: a t-test is just not applicable if you combine years with and years without SSWs, see my comment above. I think you should study the change in years with and without warmings separately; then you can provide a robust measure of the significance. Also, this would make the results more comparable to the observations shown in Seppaelae et al., 2013, for the stratospheric response, as they also analyze years without warmings.*

We followed the suggestion of the reviewer and redid Figures 2-4 from the paper separately for SSW and no-SSW (see Fig. 2-4 only for no-SSW). For each SSW event the according season was marked as "with SSW". If a SSW occurs in February or November, also the next season was marked as "with SSW" (i.e., for February MAM and for November DJF). This method ensures the consistency of each season and prevents an influence of early or late SSWs on the next season. In total, we obtained 74 (71) winters without SSW for piControl (strato-O3). Comparing Fig. 2-4 to Figures 2-4 of the paper, we see that they are very similar and our conclusions still hold. For strato-O3 the signal, especially in the late winter, even weakens. We understand that the inclusion of SSW winters in the context of EPP forcing is somewhat problematic as such events, depending on their time of occurrence, may in reality or coupled chemistry models (not in our idealized setting) influence the forcing (i.e., polar ozone depletion) itself. On the other hand, discarding SSW winters might actually remove a big part of the signal, as

a forcing may also change the timing of SSW occurrence (Gray et al., 2013). The QBO dependence of solar UV effects on the polar winter stratosphere, as shown e.g., by Labitzke et al. (2006), is strongly dependent on SSW occurrence. Additionally, see above, we don't think that the notion of a bimodal distribution of winter states is correct. Therefore, we strongly prefer to keep the figures showing all years (SSW+no- SW). But we added information on the changes in temperature and zonal wind if only no-SSW seasons are considered.

*Page 8, line 4: the impact in the winter-time high latitude upper stratosphere temperatures you show in Figure 2 has a similar structure to observed temperature and wind field changes for years with high geomagnetic activity (Lu et al., 2008; Seppaelae et al., 2013). However, the amplitude of the warming is much smaller (about one order of magnitude?) than in the observations. This comparison to observations needs to be discussed here.*

It is true that our results match qualitatively very well the results (for temperature) obtained from reanalysis data (Lu et al, 2008, Seppälä et al., 2013). Whereas for the zonal wind response, the two studies differ from each other. Seppälä et al. (2013) showed a strengthening of the polar vortex with enhanced equatorward planetary waves, whereas Lu et al. (2008) showed a weakening of the polar vortex. We added this information to the manuscript.

*Page 8, line 8: the interhemispheric coupling is evident in both the meso-O3 and the strato-O3 experiments as a "statistically significant" change in the summertime upper mesosphere. However, this is more likely an affect of SSWs?*

We checked this for only winter without a SSW and still found a warming in the summer upper mesosphere (see Fig. 2). This suggests that the signal is not an effect of

SSWs. While we find this strong change very interesting, we think that further analysis is beyond the scope of this paper.

*Page 8, line 25-30: The patterns and amplitudes you observe here should be compared to observations (Seppaelae et al., 2009; Maliniemi et al., 2014). However, as the amplitudes of your stratospheric warming appears to be much lower than observed, I would expect the impact on the troposphere also to be low compared to observations. Another point: Seppaelae et al., 2009 show that the impact on surface temperatures is different, with larger amplitudes, when years with SSWs are not considered. You should separate years with and years without warmings here as well. Can you reproduce their result regarding the impact of warmings? Again, a t-test is not applicable if you use years with and without warmings.*

We added a comparison of the surface temperature response to observations. In addition to the analysis of the full sample we analyzed the impact restricted to winters without SSW (see Fig. 4). A more detailed description on how this subset is calculated is given in the comment "Page 5-8". We obtained larger amplitudes in the surface temperature for winters without SSW. However, till much smaller than in Seppälä et al. (2009) and Baumgaertner et al. (2011). The cooling over Northern America agrees qualitatively with the aforementioned studies, but we obtained no warming over Eurasia. We added the behavior for winters without SSW to the manuscript.

*Page 11, 11: "Our results suggest that the climate impact of an ozone loss due to EPP is small" considering that the impact of particle precipitation in your analysis is masked by the strong variability implied on the Northern hemisphere winter atmosphere by sudden stratospheric warmings, and your results of the stratospheric impact strongly underestimate the observed response of the stratosphere, you can not draw this con-*

*clusion at this point.*

We rewrote the whole paragraph and encourage now more research to clarify the effects of EPP. We think it is important to point out that the climate impact of EPP is not as clear as often thought.

[Figure]

[Figure]

**Fig. 1.** PDFs for winter polar mean (left) temperature averaged over 60 – 90N and (right) zonal wind at 60N between 1 and 10 hPa. Two experiments are depicted: (black) strato-O3 and (blue) piControl.

[Figure]

**Fig. 2.** Same as Figure 2 (in paper) but only for winters without SSW.

[Figure]

**Fig. 3.** Same as Figure 3 (in paper) but only for seasons without SSW.

piControl                                    [K]          DJF: strato-O$_3$ - piControl        [K]

a)                                                        b)

**Fig. 4.** Same as Figure 4 (in paper) but only for winters without SSW.

---

## Author Comment (AC2) · 13 Oct 2017

We thank the reviewer for the assessment of our work and the useful suggestions for improvements. Below we respond point by point, first showing the reviewer's comments in blue and italic followed by our response. To avoid confusion, we refer to graphics shown in this document as Fig. and graphics shown in the paper manuscript as Figures.

*The manuscript presents the response of the atmosphere and surface temperature to*

[Figure]

*the introduced permanent decrease of the ozone concentration in the mesosphere and upper stratosphere simulated with the MPI-ESM model. The forcing was designed to mimic the ozone depletion by hydrogen and nitrogen oxides formed by the precipitating energetic particles. The subject of the manuscript is appropriate for ACP because it addresses widely discussed during the last decade question about possible influence of the energetic particles on the atmosphere, ozone and surface air temperature. The manuscript is well written, the most of relevant publications are cited, the figures are clear. However, the manuscript does not look mature because the bold conclusions cannot really be supported by the presented results. It seems obvious for the authors because in the summary they formulate why the results are not convincing and what to do to make them better. Therefore, I cannot recommend publication in the present form.*

*Main Issues:*

*1. The experimental design is too simplified. It resembles the ozone loss due to EPP obtained from the observations and models however substantially differs in the time evolution and distribution in space. Application of realistic ozone depletion scenarios could lead to very different results. If the authors do not know the implications of the chosen scenario (as it is said in the summary) what potential readers could learn from the paper? There are several aspects of the problem such as shift of the vortex from the pole and intensified ozone influence on solar radiation heating or interaction of the propagating disturbance with internal variability modes like PJO. These effects are automatically taken into account in the models considered all relevant to EPP processes, but they are missed if too simplified approach is applied. The simplest way to avoid the problem is to eliminated connection with EPP. Actually, the introduced ozone depletion scenario in the upper stratosphere is closer to the influence of halogens.*

We agree with the reviewer that our description of the experiments was too brief. However, its simplistic nature is intended and, we think, useful. We added two paragraphs to Section 2.1, also taking into account the comment of Reviewer #1. Earlier studies (see introduction for references) consider a mix of stratospheric and mesospheric ozone losses. The sole impact of a mesospheric ozone loss due to the direct EPP effect as suggested by Andersson et al. (2014) remains unclear. Additionally, a stratospheric warming due to EPP was identified in reanalysis data (Lu et al. 2008; Seppälä et al. 2013), whereas model studies obtained a stratospheric cooling either of dynamical origin (Baumgaertner et al. 2011) or of radiative origin (Arsenovic et al. 2016). In this sense, we believe that our experimental design is justified, because a) we can separate the climate impact of stratospheric and mesospheric ozone loss due to EPP; and b) the simplified approach allows us to gain insights in the processes governing the climate impact of EPP. Prescribing complex ozone reductions that vary in space, interseasonally and interannually, or simulating the ozone reduction interactively, might enable more realism but doesn't facilitate the identification of potential mechanisms. We think a reader can learn from our study that a) a significant climate impact of a mesospheric ozone change as suggested by Andersson et al. (2014) seems unlikely; and b) the interplay of dynamical cooling and radiative warming is complex and the climate impact of stratospheric ozone losses due to EPP is not as clear as often thought. In our simulations, we obtained a radiative warming in November and January. But in December, when the polar night is shortest and, hence, the radiative warming is strongest, a dynamical cooling is found. Therefore, additional research is needed to clarify the role of wave reflection for the dynamical feedback and for the coupling mechanism between stratosphere and troposphere. Furthermore, wenow added a discussion on how our experiments differ from the observational record. In particular, the lack of downward propagation of the signal, the shift of the polar vortex and the EPP restricted to the auroral oval are now discussed.

*2. I found interesting a large disagreement between the results of 80 and 150-year*

*long runs. I guess, this phenomenon should be understood and explained with more details. I am not convinced that it is just the results of inter-annual variability. If so all modeling community is in a huge trouble. Did the authors check the presence of any model drift?*

We agree with the reviewer that this large disagreement is interesting. Following your recommendation, we show different quantities that one might assume to influence EPP signals if they were drifting (Fig. 1). We do not find any drift in the model. The maximum difference (highest value – lowest value) in the sea surface temperature is 0.2 K for piControl and 0.17 K for strato-O3. This agrees with the internal variability in global mean surface temperature estimated by Sutton et al. (2015) for CMIP5 pre- ndustrial control experiments. We added a sentence to the manuscript and stated that no model drift is found.

*3. The authors frequently discuss not statistically significant responses. I have noticed that almost all results presented in Figure 2 and 4 are not significant. It is rather interesting why the applied model is not sensitive to 20% decrease of the ozone in the polar upper stratosphere. There were several publications (mentioned in the introduction) claiming significant response of the atmosphere to the observed ozone depletion in the last decades of 20th century and the ozone depletion scenario is close to what is used in the manuscript. Some discussion of this issue is necessary.*

It is true that most signals in Figures 2 and 4 are not significant at the 95Nevertheless, it makes sense to analyze if the signals could have a physical explanation and not be purely accidental. Additionally, we want to emphasize that even if DJF averages are not significant, this can be different for individual months, as we show in Figure 3. Graf et al. (2007) and Langematz et al. (2003) used observed ozone changes to analyze the role of ozone for climate change. In both studies, the ozone is mostly reduced in the lower stratosphere, in contrary to the upper stratosphere in our study. Additionally,

they used a rather short simulation period (10 years in Graf et al. (1997) and 20 years in Langematz et al. (2003)). Analyzing different simulation periods we obtain mesospheric warming and cooling of apparent significance. However, also compared to observational records of temperature and zonal wind responses due to EPP (Lu et al. (2008) and Seppälä et al. (2013)), the amplitude of our responses are smaller. We now added a comparison to the above mentioned studies.

*4. Section 3.1: The use of 75N should be better motivated if the authors would like to wire these results with ozone depletion due to EPP. If the ozone depletion occurs inside polar vortex then 75N is not representative because huge ozone influence on solar heating rate outside polar night area will dominate over very small longwave effect. It should be also considered that in the Northern hemisphere the vortex is not stable and tends to move from the pole out of the polar night area.*

Figure 1 is only an illustrative example of polar ozone heating rates and it is not thought to be representative. At other latitudes the polar night would be, of course, shorter or longer. Additionally, we agree with the reviewer that the length of the polar night exposure of an air parcel depends on altitude and the actual dynamics (e.g., movement of the air parcel). The pure radiative response to ozone loss should be a warming in mid-winter and get weaker towards early and late winter. However, our Figure 3 shows a warming in November and January/February, but not in December. The December cooling is of dynamical origin. We now discuss the missing shift of the polar vortex in Section 2.1 and added the above mentioned information to Section 2.2.

*Minor Issues:*

*1. Page 2, line 2: if –> of.* Done.

*2. Page 2, line 4: Langematz et al. (2003) showed tiny direct LW warming (Fig.7) , but the resulting stratosphere is cooler (Fig.8). Graf et al., (1998) showed the response in the lower stratosphere (70 hPa).* Thank you for pointing this out. We changed the sentence to: "During polar night reduced ozone slightly decreases the infrared cooling of the polar stratosphere resulting in a net (small) stratospheric warming (Graf et al., 1998; Langematz et al., 2003). However, both studies prescribed an ozone loss in the lower stratosphere."

*3. Page 3, line 23-25, line 31: The ozone depletion scenario is too simplified.* We extended the description of the applied ozone losses and discuss now differences to observed changes. See also reply to major comment 1.

*4. Section 2.2: The radiation code is not described. The references do not provide satisfactory information about the treatment of solar (e.g., spectral range coverage, spherical) and infrared (e.g., LTE treatment) radiation. The standard version of the RRTMG does not include wavelengths shorter 200 nm and therefore the heating rate in the mesosphere should be heavily underestimated due to the absence of Lyman-alpha line and Schumann-Runge bands. How it is treated in Psrad?* The solar and infrared radiation is treated in Psrad in the same way as in RRTMG. Hence, wavelengths shorter than 200 nm are not included. However, the absorption of ozone takes primarily place in three spectral regions: Hartley band (200 – 310 nm), Huggins band (310 – 350 nm) and Chappius band (410 – 750 nm) (Brasseur and Solomon, 2005). All of those bands are considered in RRTMG and, hence, also in Psrad. The Schumann-Runge bands are of great importance for the mesosphere, but primarily due to absorption of molecular oxygen (and not ozone). In this sense, we underestimate the total heating rate in the mesosphere. However, in our setup we compare two radiative transfer calculations. The difference between both calculations is not (at least not strongly) affected by the underestimated total heating rate. We extended the description of Psrad in the manuscript.

*5. Page 4, lines 16-18: I do not understand what means "separately . . .and then combined". Why CO2 is not in the input list. Is it not included in Psrad?* We rewrote this sentence to make it clear that the optical properties are calculated for shortwave and longwave separately, but then combined to estimate the total heating rate. CO2 and O2 are set to fixed values invariant with height in Psrad. We added this information.

*6. Page 4, line 24: Actually, the length of the polar night depends on the altitude and at 80 km it could well be shifted by one month relative to the surface. In Figure 1 this effect is absent, which affects the results in the mesosphere.* Thank you for pointing this out. Please see also comment to major point 4. We added a discussion on the representativeness of 75N to the manuscript.

*7. Page 5, line 4: The maximum of the ozone VMR is normally around 6 hPa for this location. What ozone profiles were used?* We used ozone profiles averaged over the late 20 th century provided by the general circulation and chemistry model HAMMONIA. In this profile the maximum ozone VMR is also around 6 hPa. The strongest heating occurs around the stratopause, which agrees e.g., with Brasseur and Solomon, 2005). We adjusted the sentence accordingly.

*8. Page 5, line 33: I guess, Langematz et al. (2003) showed the same.* We added Langematz et al. (2003) to the references.

*9. Page 6, line 9: 75N is not really representative (see above).* Please see the comment to major point 4.

*10. Page 8, line 5: 75N is not really representative (see above). This result disagrees with Langematz et al. (2003, see their Figure 7 and 8).* Langematz et al. (2003) showed a dynamical cooling in the polar winter stratosphere but expected also a warming from the radiative transfer modeling. In contrast, Lu et al. (2008) and Seppppälä at al. (2013)

showed a warming in the polarwinter upper stratosphere due to EPP in re-analysis data, but the magnitude is much stronger ( 5 K) than in our simulations. Furthermore, we found a small dynamical cooling in December, which is caused – as in Langematz et al. (2003) – by a reduction of waves entering the stratosphere. In this context, we discuss the differences to Langematz et al. (2003). We added the comparison to Lu et al. (2008) and Seppälä et al. (2013) to the manuscript.

*11. Page 8, line 15: statein –> state in.* Done.

Additional references:

Sutton, Rowan, Emma Suckling, and Ed Hawkins. "What Does Global Mean Temperature Tell Us about Local Climate?" Phil. Trans. R. Soc. A 373, no. 2054 (2015): 20140426. doi:10.1098/rsta.2014.0426.

Brasseur, Guy P., and Susan Solomon. Aeronomy of the Middle Atmosphere: Chemistry and Physics of the Stratosphere and Mesosphere. Springer Science & Business Media, 2005.

Lu, Hua, Mark A. Clilverd, Annika Seppälä, und Lon L. Hood. "Geomagnetic Perturbations on Stratospheric Circulation in Late Winter and Spring". Journal of Geophysical Research: Atmospheres 113, Nr. D16 (2008): D16106. doi:10.1029/2007JD008915.

Seppälä, A., H. Lu, M. A. Clilverd, und C. J. Rodger. "Geomagnetic Activity Signatures in Wintertime Stratosphere Wind, Temperature, and Wave Response". Journal of Geophysical Research: Atmospheres 118, Nr. 5 (2013): 2169–83. doi:10.1002/jgrd.50236.

[Figure]

**Fig. 1.** 10-year running mean of global SST; occurrence of SSW; 10-year running mean of global water vapour [10 hPa] and 10-year running mean of NAM index [1000 hPa] for piControl (blue) and strato-O3 (red).

---

## Author Response (AR2)

Reviewer #1:

We thank the reviewer for the positive assessment of our work. Below we reply point by point, first showing the reviewers comments in italic and blue followed by our response.

I found that my comments were considered carefully, and I recommend publication as is. I have a few minor comments which authors might consider for the final manuscript listed below.

Abstract: You could add one sentence stating the magnitude of the stratospheric ozone loss due to the indirect effect (10-15%, see line 20) observed by satellite instruments, as you did for the mesospheric (direct effect) ozone loss in line 4. Done.

Line 15: it is either "mainly depletes ozone in the mesosphere" or "depletes ozone mainly in the mesosphere" Done.

*Line 9-10: Fytterer et al only investigate the MIPAS period (2002-2012).* This is true. We rewrote the sentence.

Line 33: shouldn't that be "the 0.025 and 0.975 percentiles of the Student's t-distribution"? Done.

Line 31: changes in heating rates due "to" reduced ozone. Done.

Lines 18 and followings, discussion of differences in results: as far as I know Baumgaertner et al also did not have an interactive ocean in their model setup - can that make a difference too? In general,

we can imagine two ways how an interactive ocean can influence the results. On the one hand, an interactive ocean can be source of internal variability (obscuring the actual response). To adress the additional variability introduced by an interactive ocean, we aimed for a long simulation period. On the other hand, the absence of an interactive ocean can damp the actual response. Several publications suggest that an interactive ocean seems to be essential for the response of the solar forcing (e.g., Thiéblemont et al., 2015). We added one sentence on the differences between Baumgaertner et al. and our study to the manuscript.

Additional references:

Thiéblemont, R. *et al.* Solar forcing synchronizes decadal North Atlantic climate variability. *Nat. Commun.* 6:8268 doi: 10.1038/ncomms9268 (2015).

**Reviewer #2:**

We thank the reviewer for the assessment of our work. Below we respond point by point, first showing the reviewer's comments in blue and italic followed by our response.

The authors have not satisfactory addressed my reservations. The title is still not consistent with the subject of the manuscript. The main conclusions are not supported by the presented results. Therefore, I cannot support publication of the manuscript because it will be misleading for the community. My main reasons are the following. The applied forcing does not represent the forcing from energetic particles. The manuscript rather describes the model sensitivity to artificial ozone depletion. This problem could be eliminated by excluding energetic particles from the text, but the authors do not think it is appropriate. The obtained results (absence of atmosphere response to 20% decrease of the ozone in the stratosphere) disagree with many previous publications. For example, Langematz et al., 2003 showed significant cooling applying almost the same stratospheric ozone depletion (see Langematz et al., 2000, Figure 1b). I am not sure that the authors can properly explain their results and the manuscript obviously needs more work.

We are sorry that the reviewer is not satisfied with our response. We followed the suggestion of Reviewer #3 and modified the title to 'Climate impact of Idealized Winter Polar Mesospheric and Stratospheric Ozone Losses as caused by Energetic Particle Precipitation'. We state now clearer in abstract, introduction and conclusion that we conducted idealized model experiments. We hope that this finds the agreement of the reviewer.

Regarding the second point that our results disagree with previous studies, we believe that we have extensively discussed the difference of our study and the mentioned – previous – study in our first response. To summarize here, Graf et al. (2007) and Langematz et al. (2003) used observed ozone losses, which were applied mainly in the lower stratosphere. In contrast, our study reduces ozone mainly in the upper stratosphere. We show a similar radiative response as Langematz et al. (2003) for polar boreal winter. And again similar as they, we found a reduction of the mid-latitudinal wave flux entering the stratosphere causing a dynamically induced cooling. Additionally, both aforementioned studies used a rather short simulation period (10 years in Graf et al. (1997) and 20 years in Langematz et al. (2003)). Analysing different sub-periods of our simulation, we obtain mesospheric warming and cooling of apparent significance.

**Reviewer #3:**

We thank the reviewer for the at this late stage assessment of our work and the help to impreove the manuscript. Below we reply point by point, first showing the reviewers comments in italic and blue followed by our response.

This study employs idealized model experiments to analyze separately the impact of mesospheric and upper stratospheric ozone reductions as induced by EPP (direct and indirect effects, respectively) in boreal winter on the thermal structure and zonal winds in the middle atmosphere, as well as on surface temperatures. The topic of this paper is of high relevance since the inclusion of EPP as part of the solar forcing in the upcoming CMIP6 model experiments has recently been recommended. The choice of an idealized ozone forcing, allowing to separate the impacts of mesospheric and stratospheric ozone loss, is, in principle, a justified approach to investigate the different roles of mesospheric direct EPP impacts and stratospheric indirect effects and their underlying mechanisms, a topic that is recently widely discussed in the community. The authors have done in general a great job in responding to the comments raised by previous reviewers.

However, the suitability and limitations of the idealized experimental setup for a quantitative assessment of EPP climate impacts is to my opinion still not sufficiently discussed. The authors have chosen intentionally a simplistic experimental setup in order to be able to specifically address the different processes related to direct and indirect EPP impacts and the identification of mechanisms for possible climate responses. However, this idealized approach is less suitable for a quantitative assessment of EPP climate impacts which would clearly benefit from the use of a realistic, transient forcing in consonance with recent observations. In this sense, the focus of the paper should be clearly on the former, and this should be better reflected in the title, abstract and the conclusions. As a minimum, it should be mentioned in the title and abstract that \*idealized model experiments\* have been conducted to identify the climate impact of mesospheric and stratospheric ozone loss \*as caused\* by EPP. It should also be clearly stated in the tile and in the abstract that the study focuses on boreal winter.

We followed the reviewer's suggestion and changed the title to 'Climate impact of Idealized Winter Polar Mesospheric and Stratospheric Ozone Losses as caused by Energetic Particle Precipitation'. Furthermore, we state in the abstract, introduction and conclusion clearer that the focus of the paper is to identify the relevant processes.

Some discussion on the limitations of the chosen experimental setup has been included in the revised manuscript. However, the limitations related to the use of an ozone forcing being constant in time - to my opinion the most relevant limitation - is mentioned only marginally. EPP impacts, particularly due to the indirect effect, are mostly restricted to polar winter and it is not clear at all how the application of an unrealistic ozone forcing in polar summer could interfere with early winter EPP effects. The polar summer temperature and zonal wind responses are large (see Fig 3) and lead to a strengthening of the early polar winter vortex that could modify the early winter EPP response. The authors speculate at p4 118 (of the track change version) that "it is unlikely that signals in summer affect the climate of the next winter". However, more evidence needs to be provided to support this statement. It is further not clear to me at all what is the advantage of using an ozone forcing constant in time instead of using an annually repeating pattern. The choice of the former clearly needs to be motivated.

To check the concerns of the reviewer, we carried out additional simulations in which the ozone is only reduced from December to March. This time period was chosen, because Damiani et al. (2016) and Fytterer et al. (2015) highlighted – although for the southern hemisphere – the importance of the ozone loss in late winter. Note that the input data of ozone are monthly means and the ozone concentrations are interpolated between two subsequent months. This means a weak reduction of ozone starts already in mid-November. Figures 1 to 3 in this document resembles Figure 2 to 4 of the

paper but ozone loss applied only from December to March. Comparing the figures, we see that they are qualitatively very similar and our conclusions still hold. The magnitude is somewhat smaller than for the experiments with constant ozone loss. However, the dynamically induced cooling in December is still evident if the ozone is only reduced from December to March. We chose the ozone loss constant over time intentionally, because this allows us to investigate other seasons despite the winter. Especially, the transition seasons (spring and autumn) are of interest. We added this information to the manuscript.

Overall, I think that this paper is well suited for publication in ACP after addressing the concerns raised above, as well as some specific and minor comments listed below.

Specific and minor comments (pages and line numbering refer to the track change manuscript):

**p2 112: please use Ap instead of AP. Done.**

*p3 131-32: Here it is stated that ozone concentrations are averaged over 1850-1860, while in the response to reviewer#2 you say that you used "ozone profiles averaged over the late 20th century provided by the general circulation and chemistry model HAMMONIA". Could you please clarify?* Reviewer #2 asked about the ozone profiles of the radiative transfer model PSrad. For calculations with this model, we used ozone profiles averaged over the late 20th century provided by the general circulation and chemistry model HAMMONIA". Could you please clarify? Reviewer #2 asked about the ozone profiles of the radiative transfer model PSrad. For calculations with this model, we used ozone profiles averaged over the late 20th century provided by the general circulation and chemistry model HAMMONIA. However, for the MPI-ESM we used pre-industrial ozone concentrations averaged over 1850-1860.

*p6ff132-11: I wouldn't say that a heating response of 0.1-0.3 K/day is small. This is of the same order as what is caused by UV-induced ozone increases in the tropics. This is important since, at the end, it is the latitudinal gradient that is thought to be responsible for dynamical responses to both EPP and UV, no matter if it is introduced by a warming in the tropics or at the poles.* We agree with the reviewer that the changes in the heating rates of about 0.1 K/day (for the 20% stratospheric ozone loss) is in the range of the UV solar forcing. We modified the relevant sentence and avoided to call the heating response "small".

p13 11-3: A possible reason for the different temperature response compared to the DJF responses in previous model studies could be the use of an ozone forcing constant in time. A more realistic forcing as used in the previous studies would result in a negligible early winter (heating) response as it takes until mid winter to bring ozone-depleting NOx into the stratosphere. This should be discussed. We redid Figures 2 to 4 of the paper for simulations in which the ozone is only reduced from December to March (see Figures 1 to 3 in this document). A more detailed description of the figures is provided above. We still obtain different temperature response for DJF compared to previous papers. Also if we omit the December (cooling) response, we would still obtain a warming and not a cooling signal. We added this information to the manuscript.

*p13 113-21: The authors have added a paragraph that stresses the need for more research on the effects of EPP and its climate impact. This definitely a good point. However, they removed the sentence about the possible limitations of their analysis regarding climate impacts due to the choice of a simplified experimental design. Why?* We modified the sentence in such a way that we now state the implications of our study created by the simplified design. Before, we simply stated that the experimental design may affect the results. This was a suggestion of Reviewer #2. But we added a new paragraph to the conclusion discussing the limits of our simplified design.

*Fig 4: The caption is not consistent with the figure of the left panel (showing absolute NH surface temperatures in DJF rather than SON SH responses).* We apologize, in an earlier version of the manuscript the SON SH response was also shown. We corrected the caption.

Figure 1: Same as Figure 2 in the paper but with ozone loss only from December to March.